# Nipah Virus Efficiently Replicates in Human Smooth Muscle Cells without Cytopathic Effect

**DOI:** 10.3390/cells10061319

**Published:** 2021-05-25

**Authors:** Blair L. DeBuysscher, Dana P. Scott, Rebecca Rosenke, Victoria Wahl, Heinz Feldmann, Joseph Prescott

**Affiliations:** 1Laboratory of Virology, Division of Intramural Research, NIAID, NIH, Hamilton, MT 59840, USA; blair.debuy@gmail.com; 2Fred Hutchinson Cancer Research Center, Vaccine and Infectious Disease Division, Seattle, WA 98109-1024, USA; 3Rocky Mountain Veterinary Branch, Division of Intramural Research, NIAID, NIH, Hamilton, MT 59840, USA; dana.scott@nih.gov (D.P.S.); rebecca.rosenke@nih.gov (R.R.); 4National Biodefense Analysis and Countermeasures Center, Department of Homeland Security, Frederick, MD 21702, USA; Victoria.Wahl@ST.DHS.GOV; 5Center for Biological Threats and Special Pathogens, Robert Koch Institute, 13353 Berlin, Germany

**Keywords:** Nipah virus, endothelial cells, smooth muscle cells, henipavirus, paramyxovirus, bat virus, fusion, syncytia

## Abstract

Nipah virus (NiV) is a highly pathogenic zoonotic virus with a broad species tropism, originating in pteropid bats. Human outbreaks of NiV disease occur almost annually, often with high case-fatality rates. The specific events that lead to pathogenesis are not well defined, but the disease has both respiratory and encephalitic components, with relapsing encephalitis occurring in some cases more than a year after initial infection. Several cell types are targets of NiV, dictated by the expression of the ephrin-B2/3 ligand on the cell’s outer membrane, which interact with the NiV surface proteins. Vascular endothelial cells (ECs) are major targets of infection. Cytopathic effects (CPE), characterized by syncytia formation and cell death, and an ensuing vasculitis, are a major feature of the disease. Smooth muscle cells (SMCs) of the tunica media that line small blood vessels are infected in humans and animal models of NiV disease, although pathology or histologic changes associated with antigen-positive SMCs have not been reported. To gain an understanding of the possible contributions that SMCs might have in the development of NiV disease, we investigated the susceptibility and potential cytopathogenic changes of human SMCs to NiV infection in vitro. SMCs were permissive for NiV infection and resulted in high titers and prolonged NiV production, despite a lack of cytopathogenicity, and in the absence of detectable ephrin-B2/3. These results indicate that SMC might be important contributors to disease by producing progeny NiV during an infection, without suffering cytopathogenic consequences.

## 1. Introduction

Nipah virus (NiV) (family; *Paramyxoviridae*, genus; *henipavirus*; species *Nipah henipavirus*) was first recognized as a zoonotic pathogen in 1998–1999 in Malaysia, when it infected 265 people and caused 105 deaths. NiV was subsequently found to originate from pteropid bats (*Chiroptera: Pteropodidae*), its natural reservoir [1,2]. Since its discovery, outbreaks of NiV disease have occurred on an almost annual basis in India and Bangladesh, with resultant high case-fatality rates, often as high as 100% during small isolated outbreaks [3,4,5]. Clinically, human infection is characterized by fever, cough, dyspnea, headache, and loss of consciousness. The average duration of illness is approximately 9 days [6]. Autopsies of patients that have succumbed to the NiV infection have highlighted hallmarks of the disease, including systemic vasculitis and extensive endothelium destruction, and central nervous system (CNS) involvement [7,8]. The small blood vessels of the lungs and CNS, along with the heart and kidneys, are primary targets of infection and histologic changes. The endothelium of medium and large vessels are less involved, compared to the microvasculature, although infection of larger blood vessels can occur [9,10]. Blood vessels infected with NiV display marked inflammation, with leukocyte infiltration, thrombosis, necrosis, and often hemorrhaging is noted [8]. Multinucleated giant cells, resulting from the fusion and syncytia formation of infected endothelial cells (ECs) are a prominent disease feature and are directly involved in the development of vasculitis [11,12,13,14]. Syncytium involving ECs are observed in alveolar spaces, causing pulmonary edema, and viral antigen is seen in cells in the endothelium, tunica media, and in the alveolar spaces [15].

The tropism of NiV is thought to be primarily dictated by the expression of ephrin-B2 and ephrin-B3 on cells, the only identified receptors for NiV [16,17,18]. These receptors are highly conserved across Mammalia, likely accounting for the broad species tropism of NiV. These receptors normally function in cell-cell signaling, angiogenesis, and neuronal axon guidance [19,20,21]. Ephrin-B2 expression is observed in arterial ECs and neurons, with a high expression in the lung and brain, while ephrin-B3 expression is restricted to the brain stem and heart [9,16,17,18,22].

In vitro, various cell lines and primary cells have been evaluated for NiV permissibility, and some for their ephrin-B2/3 expression. Susceptibility to NiV infection has been linked to ephrin-B2 or B3 expression, and even in cases where cytoplasmic entry of NiV through macropinocytosis was noted, ephrin-B2 was still required for cytoplasmic entry [23]. In vivo, it has been described that smooth muscle cells (SMCs) of the tunica media often contain the NiV antigen, but little or no pathology has been associated directly with SMCs, unlike the extensive cytopathology observed in the endothelium. SMCs form the tunica media and are tightly linked physiologically to the endothelium for control of vascular stability and function during and after blood vessel formation. SMCs and ECs are developmentally linked as well, with both originating from the lateral plate mesoderm. Although ECs express various ephrins and Eph receptors in abundance, SMCs also express ephrins and their Eph receptors spatiotemporally, and receptor and membrane-bound ligand interactions between these two cell types are important for adhesion and cell motility during angiogenesis [24].

To gain an understanding of how these two in vivo targets for infection might contribute to NiV replication and cytopathogenicity, we investigated the susceptibility of human primary SMCs to NiV infection, and the consequence of exposure to NiV, compared to ECs in vitro. We show that human SMCs are permissive for infection and produce high and sustained titers of NiV, without the cytopathic effects that are prominent in ECs, such as fusion, syncytia formation, and cell death. Ephrin-B2/B3 was not detectable on the cell surface of SMCs, however, upon exogenous expression of ephrin-B2, NiV infected SMCs gained the ability to fuse and formed syncytia similar to what was observed in EC cultures. Infection in vitro mimics the in vivo observations, indicating that SMCs contribute to the NiV disease by harboring NiV, allowing replication and the production of high amounts of progeny virus, without suffering cytopathic damage.

## 2. Materials and Methods

### 2.1. Ethics Statement

Work with NiV and all potentially infectious material was performed in the BSL4 facility at the Rocky Mountain Laboratories (RML), National Institute of Allergy and Infectious Diseases, National Institutes of Health. The Institutional Biosafety Committee (IBC) approved all of the procedures. This study used tissues from Syrian hamsters and African green monkeys that were enrolled in previously published NiV studies [15,25]. These studies were approved by the Institutional Animal Care and Use Committee and performed in accordance with the guidelines of the Association for Assessment and Accreditation of Laboratory Animal Care (AAALAC, Frederick, MD, USA) in an AAALAC-approved facility. Primary human cells for this study were obtained from a commercial source (Lonza, Walkersville, MD, USA) from anonymous donors.

### 2.2. Cells and Viruses

Vero C1008 cells were obtained from the European Collection of Cell Cultures (Salisbury, UK). Primary human lung microvascular endothelial cells (ECs, CC-2527) and primary smooth muscle cells (SMCs, CC-2581) (Lonza, Walkersville, MD, USA) obtained from human pulmonary arteries were propagated and maintained in specialized media supplemented with growth factors according to the manufacturer’s instructions. Experiments are representative of results obtained from two separate human donors for each primary cell type. Although not every experiment was performed using cells from both donors, infection kinetics and the initial fusion experiments were performed with cells from both donors as biological replicates and the results were consistent with no notable donor variation. HeLa cells stably expressing ephrin-B2 or B3 were kindly provided by Dr. Christopher Broder (Uniformed Services University, MD, USA). The Malaysian strain of NiV (NiV-M) was provided by the Special Pathogens Branch of the Centers for Disease Control and Prevention, Atlanta, GA, USA and propagated on Vero E6 cells grown in Dulbecco’s Modified Eagle’s medium (DMEM) supplemented with 10% fetal calf serum (FCS), 2 mM L-glutamine, 50 IU/mL penicillin, and 50 µg/mL streptomycin (Life Technologies, Carlsbad, CA, USA). Supernatants were clarified by low-speed centrifugation and stocks were stored in liquid nitrogen.

### 2.3. Histology

Tissue blocks from NiV-infected hamsters [15] and African green monkeys (AGMs) [25] from prior studies at RML that were removed from the BSL4 laboratory according to IBC-approved protocols were used to prepare slides for staining to assess viral tropism. Embedded tissues were processed using a Discovery XT automated processor (Ventana Medical Systems, Oro Valley, AZ, USA) with a DAPMap kit, and stained with hematoxylin and eosin (H&E). Immunohistochemistry (IHC) was performed to detect NiV antigens using a rabbit anti-NiV nucleocapsid (NP) primary antibody at 1:5000 dilution as previously described [26]. Tissues were also stained with a monoclonal mouse anti-smooth muscle actin antibody at 1:100 (Millipore) and a mouse anti-CD31 antibody at 1:700 (LifeSpan BioSciences, Seattle, WA, USA) for identification of SMCs and ECs, respectively.

### 2.4. In Vitro Infections

Cells were grown to 90–95% confluency in 48-well plates in their respective growth media. The media was then removed and 200 µL of NiV diluted in fresh media was added to the cells at the indicated multiplicity of infections (MOIs). After 1 h, the inoculum was removed, cells were washed in DPBS, and 500 µL of fresh growth media was added to the monolayer. Supernatants were collected by removing and replacing half of the media at the indicated time points until 6 days post-infection (DPI), then a complete media replacement was performed every second day thereafter. In parallel, cells identically treated were stained with a Kwik-Diff kit (Thermo Scientific, Waltham, MA, USA) to visualize the cells and assess their morphology and syncytia formation. Images were captured using a Nikon DS-Fi1 camera. For immunofluorescence assays, cells were grown in 8-well plastic chamber slides (Nunc Lab-Tek) and infected with NiV at a MOI of 5. These cells were fixed in 10% formalin overnight at the indicated time points, prior to removal from the BSL4 following IBC-approved protocols, and then stained.

For lentivirus transductions, monolayers of SMCs or ECs were exposed to lentiviruses encoding red fluorescent protein (RFP) or green fluorescent protein (GFP) according to the manufacturer’s instructions (Cellomics Technology, Halethorpe, MD, USA). Briefly, cells were incubated in growth media with 6 µg polybrene and the supplied lentivirus stock for 24 h at 37 °C, 5% CO_2_. The virus was then replaced with a growth medium. Following visualization of RFP/GFP expression in 95–100% of cells after 24–48 h, cells were sub-cultured at a 1:1 ratio. The following day, the co-cultures were infected with NiV at a MOI of 5 and visualized by either staining with Kwik-Diff stain and light microscopy or stained for NiV antigen and visualized using confocal microscopy (IFA) as described below.

### 2.5. Microscopy

Formalin-fixed monolayers of NiV-infected cells grown in 8-well chamber slides were washed in DPBS prior to incubation in 0.2% Triton X-100 for 7 min, followed by blocking in 4% BSA/PBS for 10 min. Slides were then incubated with NiV-specific rabbit antisera, washed twice, then incubated with an anti-rabbit Alexa Fluor 488 antibody (Life Technologies). After washing twice, slides were mounted using ProLong Antifade containing DAPI (Invitrogen, Carlsbad, CA, USA) and visualized by confocal microscopy.

### 2.6. Virus Quantitation

The tissue culture infectious dose 50% (TCID_50_) method was used to titrate NiV from the supernatants of infected cells as described previously [27]. Briefly, Vero E6 monolayers grown in 96-well plates were inoculated in triplicate with 100 µL of serial dilutions of supernatants in DMEM supplemented with 2% FCS. After 4 to 5 days of incubation at 37 °C, 5% CO_2_, wells were examined for cytopathic effect (CPE) and the Spearman-Karber method was used to calculate TCID_50_ values.

### 2.7. Ephrin-B2/3 Expression and Transfections

Endogenous expression of ephrin-B2 or B3 by cells was quantified by flow cytometry. Cells were collected using 100 mM EDTA and gentle scraping, washed in PBS containing 15 mM EDTA, and incubated with recombinant human EphB4, the ligand for ephrin-B2/3, conjugated to human FC (R&D Biosystems, Minneapolis, MN, USA) for 1 h. After washing in 15 mM EDTA, cells were stained with an anti-human FC antibody conjugated to Alexa Fluor 647 (Life Technologies, Carlsbad, CA, USA) and fixed in 4% PFA. HeLa cells stably expressing ephrin-B2 or B3 were used as controls. Flow cytometry was performed using a LSR II cytometer (BD Biosciences, Franklin Lakes, NJ, USA) and data were analyzed using the FlowJo software (Treestar Inc, Ashland, OR, USA).

Transfection of SMCs with human ephrin-B2 driven by a CMV promotor expression plasmid (Sino Biological Inc., Chesterbrook, PA, USA) or plasmids encoding the fusion protein (F) and glycoprotein (G) of NiV was performed using a Nucleofector kit for primary SMC transfection (Lonza, Walkersville, MD, USA). The NiV F and G expression plasmids were described previously [28]. Cells (5 × 10^5^) in the Nucleofector solution were mixed with plasmid DNA (1 µg of ephrin or 1 µg each of F and G plasmids) and loaded into a cuvette. The A033 program was used, after which the medium was added and cells were plated in 12-well plates. In the case of NiV F and G plasmid transfection, SMCs were co-cultured with ECs the day after transfection, then stained with Kwik-Diff 2 days later, and assessed for syncytia formation using light microscopy. Two days after transfection with plasmids encoding ephrin-B2, SMCs were infected with NiV at a MOI of 5 and then stained with Kwik-Diff and assessed for syncytia formation.

### 2.8. Statistics

Data obtained from the titration of NiV were plotted as a geometric mean with geometric standard deviation, using the Prism (Graphpad, v9) software (https://www.graphpad.com/support/faq/prism-900-release-notes/).

## 3. Results

### 3.1. NiV Infects SMCs and ECs of the Lung of Animal Models of NiV Disease with Distinct Histological Consequences

The disease caused by NiV in Syrian hamsters and African green monkeys (AGM), such as in humans, is characterized by systemic infection and vasculitis [29,30]. In an effort to better understand the cellular targets of NiV across species that are models for NiV disease, we focused on examining the vasculature of the lung, a major target organ affected, and leading to the respiratory component of NiV disease. The lung tissue from NiV-inoculated hamsters necropsied at 5 DPI, and AGMs at 10 DPI, were stained for NiV NP antigen, and co-stained with markers for ECs (CD31) and SMCs (smooth muscle actin). Multiple lung samples from both hamsters and AGMs showed NiV-positive staining surrounding many small arteries (Figure 1). A closer examination revealed NiV-positive cells within the SMC layer, comprising the tunica media, as well as more abundant NiV-positive cells within the endothelium. Pathologic changes in these tissues were observed in the endothelium, with syncytia formation being a common finding. Conversely, although SMCs of the tunica media contained NiV-positive cells, there were no observable pathologic changes in either hamsters or AGMs, a similar observation to what has been reported in human tissues.

### 3.2. NiV Productively Infects Human SMCs and ECs, Resulting in Disparate Cytopathogenicity

To determine the susceptibility of human lung primary SMCs and ECs to the NiV infection in vitro, we cultured low-passage (1–2 passages) lung SMCs and ECs with varying amounts of NiV, ranging from a MOI of 0.1–5. The cells were monitored for CPE and fusion events or other morphologic changes throughout the experiment. As early as 1 DPI, ECs began to display cytopathic changes, including fusion and the formation of syncytia, when infected with a MOI of 0.1 (Figure 2A, top). By 3 DPI, all of the EC monolayers had developed extensive CPE and only sparse adherent cells were present. Conversely, SMCs exposed to NiV at the same MOI (0.1) showed no cytopathic changes out to 5 DPI (Figure 2A, bottom) or even through 21 DPI (data not shown).

To assess whether SMCs exposed to NiV can be productively infected, as observed in vivo, we compared the production of progeny NiV between SMC and ECs by inoculating monolayers of each cell type with NiV at MOIs of 0.1 or 5. In both cases, peak titers of NiV were reached more quickly in EC cultures than in SMCs (2 days sooner for both MOIs), although ultimately, SMCs produced approximately the same peak titer of progeny virus (Figure 2B). The EC cultures developed extensive CPE, thus the elevated viral titers were only sustained for 2 days at the lower MOI, and 1 day at the higher MOI, and titers dropped the following day, after which the sampling of these cultures was discontinued due to the lack of viable cells. In contrast, the SMCs continuously produced between 1 × 10^4^–5 × 10^5^ TCID_50_/mL out to 21 DPI when inoculated with a MOI of 0.1, and up to 1 × 10^7^ TCID_50_/mL at early time points. After approximately 10 DPI, titers were similar in the SMC cultures, regardless of inoculum, out to 21 DPI. These results were consistent between two independent experiments using two donors of SMCs, designated 2011 SM and 2013 SM. These peak titers are comparable to levels often obtained during NiV propagation on Vero E6 cells (Figure 2B).

### 3.3. Cell-to-Cell Spread of NiV in SMC Cultures Is Limited, Compared to ECs

Since we did not observe CPE in the NiV-infected SMC cultures, we investigated the initial infection dynamics and the potential of SMCs to fuse, facilitating cell-to-cell spread of NiV. We performed an immunofluorescence time course experiment to directly compare the infection dynamics in SMC and EC cultures. Monolayers of each cell type were exposed to NiV at a MOI of 5 and stained for NiV NP over the course of 2.5 days, at matched time points. The NP of NiV was first detected in ECs 8 h post-infection, and in SMCs 2 h thereafter (Figure 3). By 12 h, ECs showed evidence of the initiation of fusion, followed by large syncytia and infection of 100% of the monolayer by 14 h. A complete destruction of the monolayer was achieved 2–4 h later. In contrast to ECs, infection of SMCs remained focal, with no syncytia formation observed, and no evidence of cell-to-cell spread at any time point. Continuing the experiment until 60 h, many cells remained uninfected, as indicated by the absence of NP staining. This result indicates that not only do SMCs not fuse or form syncytia, only a portion of the cells become infected, despite a high MOI and the continuous presence of infectious progeny virus in the supernatant, as shown in Figure 2B.

### 3.4. Ephrin-B2/3 Cell Surface Expression Is Undetectable on Primary SMCs

Since the initial infection of NiV in SMCs is inefficient and there is no cell-to-cell fusion following infection, even at a high MOI, we hypothesized that SMCs express little or no ephrin-B2 or B3. Ephrin-B2/3 is the receptor ligand identified for NiV, and is thought to be required for fusion between infected cells expressing the F and G proteins of NiV, and neighboring cells that express ephrin-B2 or B3 on their cell surface. SMCs were incubated with EphB4, the receptor for the ephrin-B2/3 ligand, fused to human FC, as a means to measure the expression of ephrin-B2/3 by flow cytometry. Analysis of both Vero E6 cells and ECs, which are susceptible to NiV-induced fusion, showed measurable amounts of ephrins on their cell surfaces, although not as much as HeLa cells that stably express ephrin-B2 or B3, used as positive controls (Figure 4). In contrast, SMCs were absent for ephrin-B2 or B3.

### 3.5. Exogenous Expression of Ephrin-B2 in SMCs Permits NiV-Induced Fusion

The lack or low level of ephrin-B2/B3 expression might prevent cell-to-cell fusion and explain the lack of pathology associated with these cells in vivo. To assess whether exogenous expression of ephrin-B2 might render SMCs fusogenic upon NiV infection, we transfected SMC with a plasmid driving the expression of human ephrin-B2 or a control plasmid expressing GFP, prior to the NiV infection. Expression of ephrin-B2 alone did not result in morphological changes in SMCs cultured in a monolayer. However, infection of ephrin-B2-expressing SMCs with NiV resulted in conspicuous syncytia formation (Figure 5 bottom) by 48 h post-infection, similar to what we observe in EC cultures. This demonstrated that NiV-induced cytopathology can occur in SMC when they express ephrin-B2, and that SMCs possess the machinery to fuse, with the exception of a cell surface receptor that interacts with the G protein on neighboring NiV-infected cells, such as ephrin-B2.

### 3.6. SMCs Expressing NiV F and G Fuse with Ephrin-B2-Expressing ECs

To further examine whether SMCs are biologically capable of fusing in the context of NiV infection, either with themselves or with ECs, we transfected SMC with plasmids encoding the F and G proteins of NiV to mimic NiV infection, with the notion that SMCs might acquire the ability to fuse with ECs, which endogenously express ephrin-B2 on their surface. SMCs expressing NiV F and G did not self-fuse (Figure 6A), showing again that these cells lack a surface receptor, possibly ephrin-B2, that can interact with F and/or G complexes to initiate fusion. However, when F and G-expressing SMCs were co-cultured with ECs, extensive fusion and syncytia formation was readily observed (Figure 6B), demonstrating that surface expression of F and G on SMCs can interact with an EC-derived surface receptor, most likely ephrin-B2, leading to efficient fusion.

Then, we used an additional approach to confirm the fusogenic capacity of SMC-ECs. We co-cultured SMCs that expressed RFP via lentivirus transduction, with ECs transduced with a GFP-expressing lentivirus, in order to distinguish between the different cell types. Uninfected cultures showed no fusion events, however, following NiV infection, fusion events between SMC and ECs were observed by visualizing syncytia that contained colocalized RFP and GFP, as well as the NiV NP antigen (Figure 6C,D) or sometimes only GFP and NiV NP, demonstrating EC-EC fusion events. These experiments show that SMCs are capable of fusing with cells that express ephrin-B2 on their surface, when NiV F and G are present on the membrane of SMCs, either by transfection or infection with NiV.

## 4. Discussion

The basic understanding of NiV pathology in humans is solely derived from histological observations from relatively few autopsies, and most of what is known overall has been gleaned from animal studies that model the NiV disease. Two prominent animal models that recapitulate many aspects of human NiV disease are the Syrian hamster, and the African green monkey [29,30]. A hallmark of NiV disease in humans and models, histologically, is the severe vasculitis observed, which is associated with NiV antigen-positive vasculature [6,14]. ECs, which line the lumen of blood vessels, as well as cells of the tunica media are sites of viral replication [6,9,28,30,31]. Infection of ECs results in the formation of multinucleated syncytia and overt disruption of the involved and neighboring cells, which is recapitulated in vitro [9]. However, little is known regarding how cells of the tunica media, SMCs, might contribute to the disease.

In vivo, we observed the NiV antigen systematically throughout the vasculature of infected animal models (hamsters and African green monkeys). Examination of lung tissues from infected animals showed numerous antigen-positive ECs, and although more rare, NiV-positive SMCs. Pathogenic changes, such as fusion and syncytia formation, were readily observed and uniquely associated with the infected endothelium. Identification of SMCs by their expression of smooth muscle actin showed that several of these cells were also NiV antigen positive, often in areas proximal to the infected ECs, and sometimes in areas where the infected ECs were not detected. Moreover, isolated infected SMCs were not associated with syncytia formation or any other overt cytopathic changes [31].

These disparate observations in histologic changes between infected ECs and SMCs were recapitulated in vitro. We used primary ECs and SMCs isolated from human lung tissue to model the susceptibility and consequences of NiV infection in an attempt to expand upon in vivo observations. Both cell types were permissive for NiV infection, however cytopathology was only observed in EC cultures. In contrast, SMC cultures remained unaffected for the duration of the study, up to 3 weeks, even with a continuous production of infectious NiV. ECs showed widespread fusion, resulting in multinucleated cells, as early as 1 DPI, with a complete destruction of the cell monolayer within 2–3 DPI. Other viruses, such as encephalomyocarditis virus, cytomegalovirus, and Epstein-Barr virus infect SMCs [32,33,34,35,36]. Infection of SMC by Epstein-Barr and encephalomyocarditis viruses leads to a lytic infection. In contrast, infection of SMCs by cytomegaloviruses leads to vial latency or persistence similar to what we observe herein [37].

We used several approaches to investigate the consequences and dynamics of NiV infection of SMCs, compared to ECs. Whereas ECs exhibited nearly 100% infection in a monolayer at a high MOI, only approximately 10–20% of SMCs became infected, even with a MOI as high as 5, and even after several days of infection. This lack of susceptibility of all cells might indicate that these cultures of primary cells contain cells in various states in their cell cycle or slight differences in receptor expression. Regardless of the fraction of SMCs that are susceptible to NiV, we were unable to detect ephrin-B2 or B3, the known entry receptor ligand for NiV, on the cell surface of SMCs. It is possible that very low levels of ephrins are expressed in these specific primary cells, accounting for the lack of complete monolayer infection. This is unlikely, however, because even after multiple days of infection, with a constant production of NiV progeny in the supernatant of these cultures, only a limited number of cells became persistently infected and no fusion was observed at any time point, unless ephrin-B2 was exogenously expressed. Following ephrin-B2 transfection, SMCs readily fused upon infection with NiV. These observations suggest that an unidentified entry receptor exists or entry in SMCs can occur by a non-specific mechanism.

Likewise, ECs, which readily fused to each other following the NiV infection, were able to fuse with SMCs only after the SMCs were transfected with plasmids that encode the F and G proteins of NiV or after infection with NiV. Taken together, these data demonstrate that SMCs lack a membrane-bound receptor, such as ephrin-B2/B3, that can interact with the neighboring infected cells, but when F and G is expressed (by transfection or infection) it can interact with the adjacent cells that express a membrane-bound receptor, resulting in fusion. Early studies aimed at identifying NiV entry mechanisms identified micropinocytosis as a possible mechanism, however, ephrin-B2 was still utilized in that instance [23]. It is possible that ephrin-B2/B3 is expressed in these SMCs, but not at the cell surface, and binding and internalization via another attachment receptor or non-specific attachment at the surface might trigger endocytosis and downstream interactions between ephrin-B2/3 in endosomes and the NiV surface proteins. This would prevent fusion between neighboring cells, yet allow for NiV entry. SMCs are known to express ephrins and Eph receptors, but expression is highly spatiotemporal [20,24]. Expression is associated with angiogenesis and development, and it is likely that SMCs of the tunica media express highly variable amounts of ephrins in vivo.

From these in vitro experiments, we can surmise that the CPE observed in ECs is almost entirely a result of fusion, as infected SMCs remained intact for several weeks following the infection. This is not surprising and much of the vasculitis observed in vivo is likely due to the direct effects of the NiV infection, and not entirely dependent on inflammation and immune cell infiltration. An examination of NiV-infected Syrian hamster tissue at very early time points shows infection of the tunica media of larger arterial vessels in the absence of infected endothelium [31]. In this case, it was hypothesized that SMCs play a role in NiV spread from the initially infected epithelial cells to the tunica media, and later to the endothelium, which then leads to syncytia formation within the endothelium and severe disease [31]. This corresponds with the low viremia associated with NiV infections in animal models, in contrast with high viremia observed for many hemorrhagic fever-causing viruses. For NiV infection, it is likely that viral particles gain access to ECs from interstitial tissues, traversing the tunica adventitia and tunica media, as opposed to the direct infection from the vessel lumen, where ECs would be the first cells to become infected. In human tissues assessed post-mortem, there is extensive infection of the endothelium with syncytia formation, as well as adjacent infected SMCs [13]. Together, these observations suggest that SMCs might act as an intermediate between the early-infected parenchyma and terminally infected endothelium. Although infection of SMCs alone is not cytopathic, SMCs likely facilitate pathogenesis by providing an amplifying medium for progeny NiV production and transmission to ECs, without suffering the negative cytopathic consequences seen in the endothelium. Currently, the specific cellular response to the NiV infection in SMCs, and how that response might be involved in the vasculitis-associated pathogenesis independent of cytopathic effects is completely uncharacterized. Future studies should be aimed at determining the entry mechanisms of NiV in SMCs, and elucidating how the response to infection of SMC might contribute to the pathogenesis of the NiV disease.

## Figures and Tables

**Figure 1 cells-10-01319-f001:**
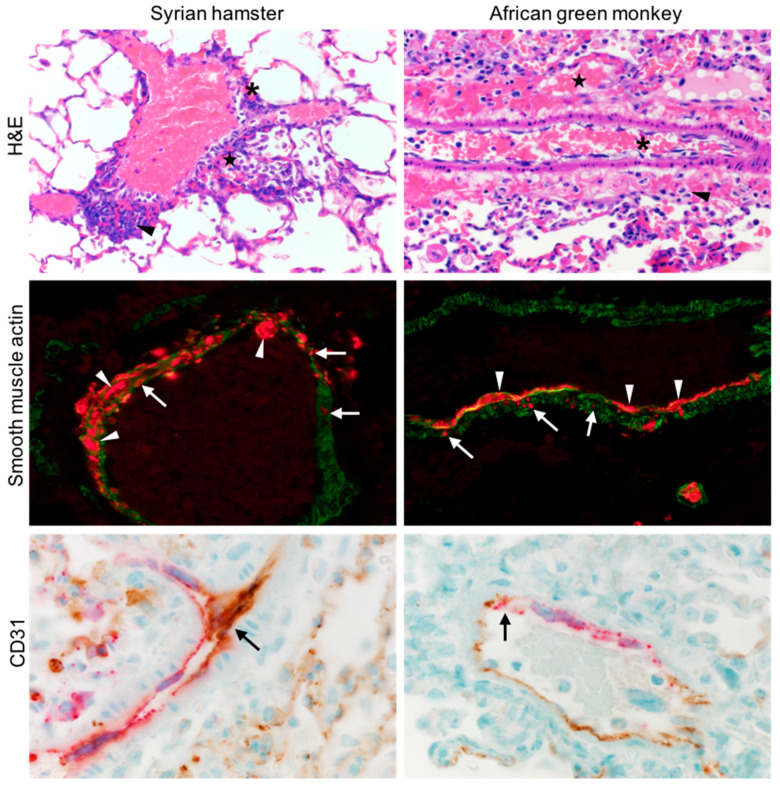
Lung tissue from hamsters (**left column**) and AGMs (**right column**) infected with NiV were sectioned and stained by H&E (**top row**). Asterisks denote degeneration of the endothelium, stars denote hemorrhage, and arrowheads highlight perivascular inflammation. NiV NP (**red**) and smooth muscle actin (**green**) are shown (**middle row**), arrows point to infected SMC, and arrowheads infected ECs, displaying syncytia. NiV NP (**red**) and CD31 (**brown**) are shown (**bottom row**), showing extensive colocalization and infection of ECs and the formation of multinucleated giant cells (**arrows**).

**Figure 2 cells-10-01319-f002:**
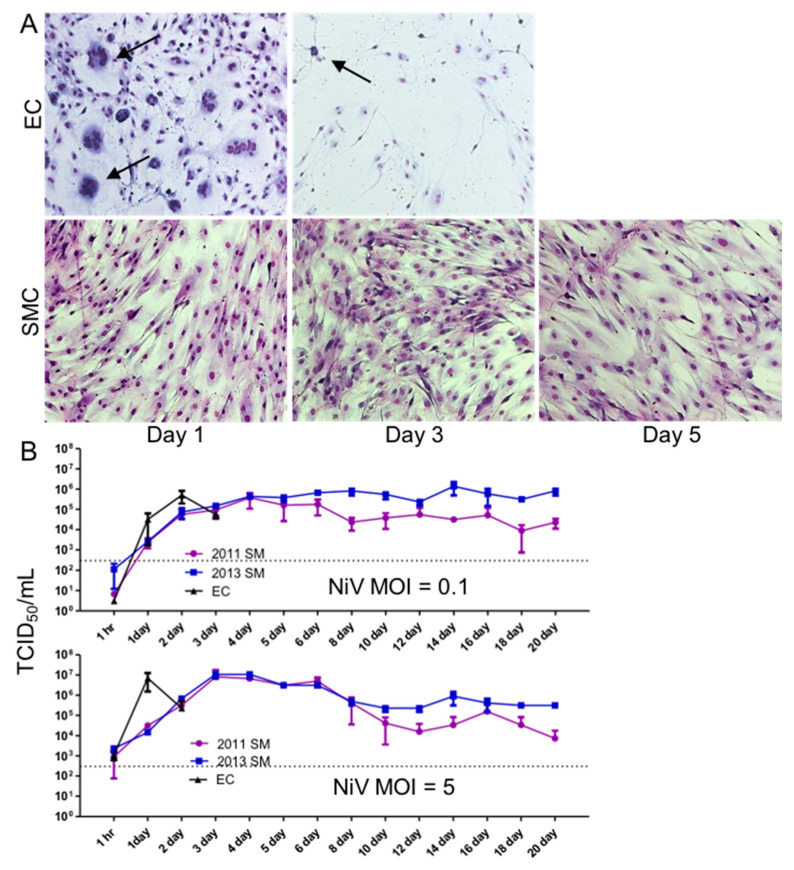
Cytopathic effect and replication of NiV in EC and SMC. (**A**). Monolayers of ECs and SMCs were infected with NiV at a MOI of 0.1 and cells were fixed and stained with Kwik-Diff for visualization. Arrows point to syncytia formation. Images were captured at 10× magnification. (**B**). ECs and SMCs were infected with NiV at a MOI of 0.1 or 5 and supernatants were collected at the indicated time points, and NiV was quantitated by titration using a TCID_50_ assay.

**Figure 3 cells-10-01319-f003:**
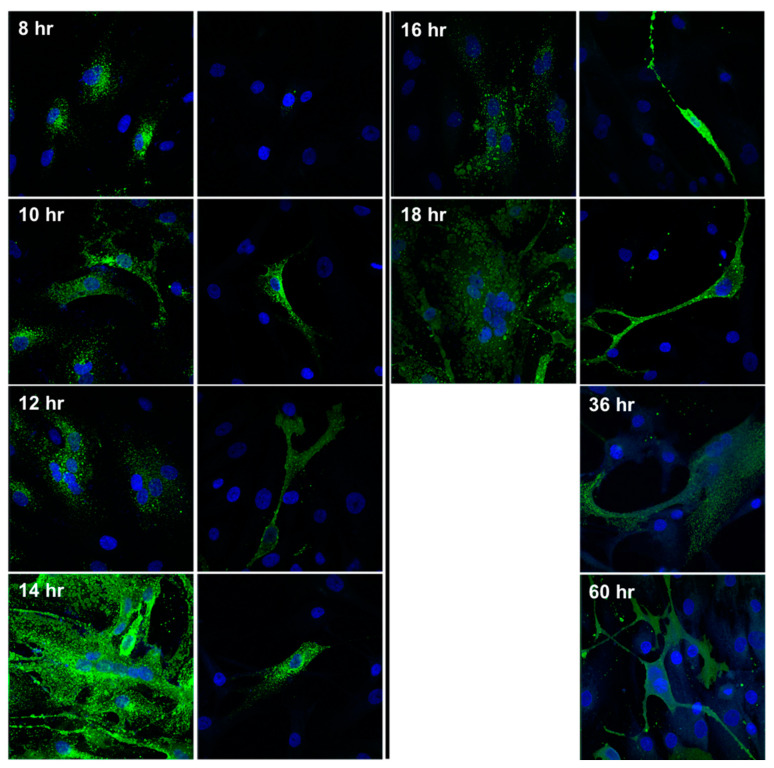
Visualization of cytopathic changes and NiV antigen in ECs and SMCs. Monolayers of the respective cell types were exposed to NiV at a MOI of 5. At the indicated time points, samples were fixed, permeabilized, and stained for NiV NP expression (green), and DAPI for nuclei staining (blue). Images were captured at 20× magnification.

**Figure 4 cells-10-01319-f004:**
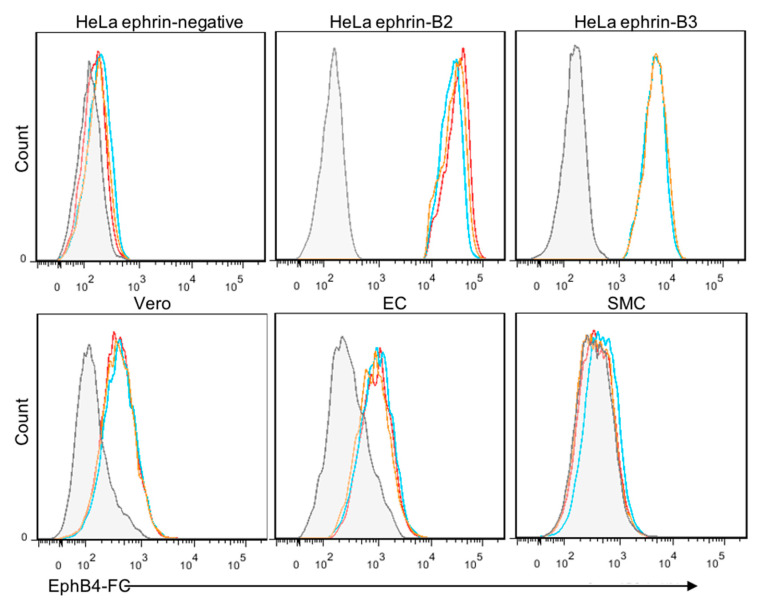
Expression of ephrin-B2/3 on ECs and SMCs. The respective cells were surface stained using recombinant EphB4 (the ligand for ephrin-B2/3) fused to human FC receptor (EphB4-FC), followed by an Alexa 647-conjugated anti-FC secondary antibody. Flow cytometry was performed and grey histograms show negative controls and colored lines represent stained cells from three independent experiments. HeLa cells were used as a negative control, and HeLa cells that stably express ephrin-B2 or B3 were used as positive controls for the assay.

**Figure 5 cells-10-01319-f005:**
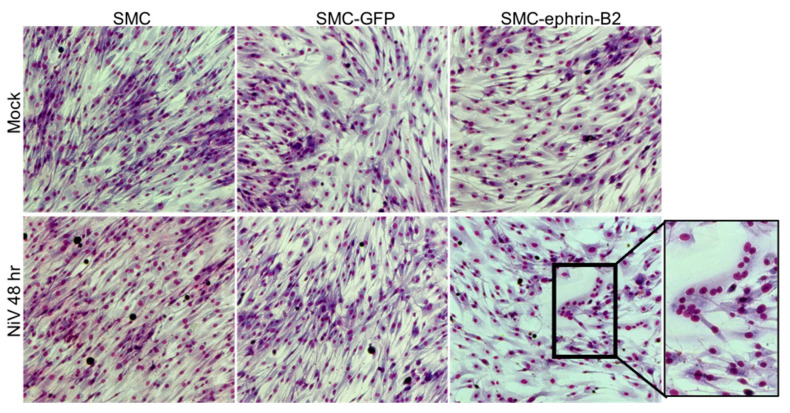
SMC transfected with a plasmid encoding ephrin-B2 can fuse following infection with NiV. SMCs were either mock transfected or transfected with plasmids expression either GFP or ephrin-B2. Following transfection, cells were exposed to NiV at a MOI of 5 or mock-infected (top row). At 48 h, cells were fixed with Kwik-Diff for visualization. Insets highlight morphologic changes. Images were captured at 10× magnification.

**Figure 6 cells-10-01319-f006:**
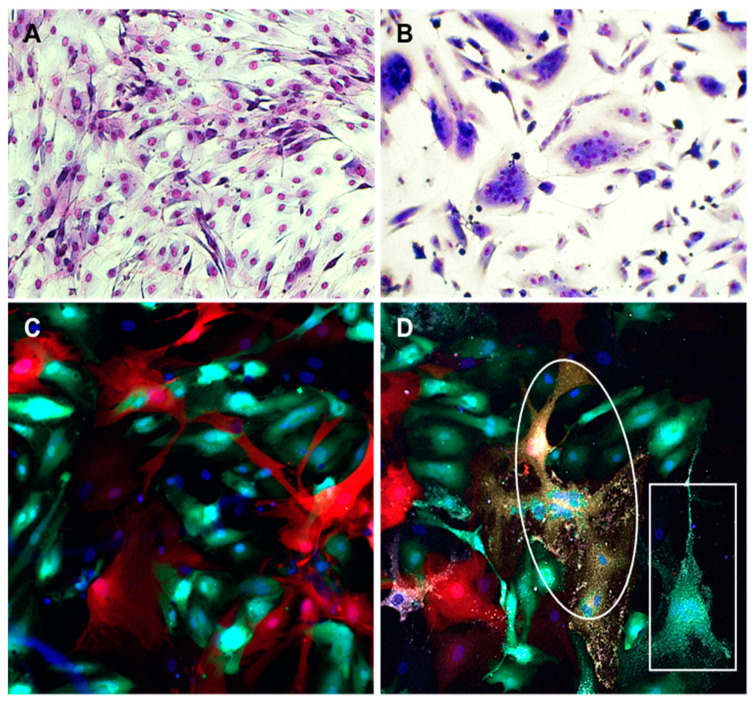
Fusion between SMC expressing NiV F and G and ECs. SMCs were transfected with plasmids encoding NiV F and G proteins and seeded in culture plates alone (**A**) or mixed with primary ECs (1:1) (**B**). Panels C and D show visualization of virus infection and syncytia in mixed cultures of SMCs and ECs. SMC and EC were transduced with lentivirus constructs expressing either RFP or GFP, respectively. After fluorescent protein expression was observed in almost all cells, SMCs and ECs were co-cultured for 24 h (1:1) and then either mock-infected (**C**) or infected with NiV at a MOI of 5 (**D**). Cells were fixed 18 h after infection and stained with anti-NiV NP antibody (white) and DAPI (blue). Syncytia were composed of colocalized RFP and GFP (oval), indicating that both SMCs and ECs were involved in fusion or only GFP (rectangle) showing fusion of ECs alone. All fusions contained NiV NP (white). Images were captured at 10× magnification (**A**,**B**) or 20× magnification (**C**,**D**).

## Data Availability

Not applicable.

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
