# Peer review of "Nipah Virus Efficiently Replicates in Human Smooth Muscle Cells without Cytopathic Effect"

_cells, 2021, doi:10.3390/cells10061319_

Round 1

Reviewer 1 Report

In this manuscript, the authors describe an interesting phenomenon regarding Nipah virus infection of the smooth muscle cells, notably the absence of cell-to-cell fusion in homogenous SMC cell cultures. Since the lack of fusion allows for the extended viability of a Nipah virus infected SMC (as compared to an endothelial cell), the authors suggest that SMCs may serve as a an important source of progeny virus- which would correlate with prior pathological findings showing infection of the tunica media. 

Overall, this manuscript is very well executed- the logical flow of experiments and their rationale was very easy to follow, and the results are clearly delineated. 

The reviewer just has a few minor comments and suggestions for the reviewers to address: 

1) Given the inability to detect surface expression of EphrinB2/B3 on SMCs, did the authors consider attempting intracellular staining to determine the presence of potential endosomal EphrinB2/B3 expression? 

2) Despite of the lack of surface Ephrin B2/B3 expression, did the authors consider analyzing levels of Ephrin B2/B3 mRNA transcripts in SMCs?  It may be the expression of certain variant forms of EphrinB2 (there are at least 4 documented variants as per EFNB2 ephrin B2 [Homo sapiens (human)] - Gene - NCBI (nih.gov)) that contribute to the observed SMC phenotype. Even conducting a simple comparison of mRNA transcript levels of EphB2/B3 between SMCs and ECs would make this work more complete. 

3) Related to comment #2- Since the authors mentioned how only a low percentage of SMC's were productively infected with NiV even in spite of high MOI infection, a relatively simple way to answer the question of mRNA transcript levels is to utilize single-cell sequencing services (e.g. 10X genomics) to analyze the EphrinB2/B3 expression patterns across a population of SMCs and see whether it correlates with the observed levels of NiV infection. 

Reviewer 2 Report

In the study “Nipah virus efficiently replicates in human smooth muscle cells without cytopathic effect”, DeBuysscher et al., present their findings on susceptibility of human smooth muscle cells to Nipah virus infection in vitro and speculate on their relevance for NiV disease in humans or animal hosts. Vascular endothelial cells are primary targets of NiV infection and display strong cytopathic effect (CPE) characterized by syncytium formation. Smooth muscle cells have been found NiV-positive in immunohistochemistry; however, no CPE has been described. To understand the contribution of smooth muscle cells to NiV pathogenesis, authors analysed susceptibility of cells to NiV, expression of ephrin-B2 and -B3 as relevant NiV entry receptors and fusogenic potential of cells after expression of NiV F and G and ephrin-B2. Overall, they concluded that smooth muscle cells might contribute to NiV pathogenesis by continuously producing progeny virus without leading to cytopathic effects.

In general, the study adds interesting data towards the understanding of NiV pathogenesis in vivo. However, the manuscript needs some additional experimental data, adjustments and corrections as well as thorough proofreading before suitable for publication:

Major points:

Authors claim that they used “several approaches to identify the mechanisms and dynamics of NiV infection of SMCs” (l. 370). Clearly, authors describe a phenotype and recover the phenotype by overexpression of the NiV entry receptor, but this is not the identification of a mechanisms or dynamics. So, please rephrase or include data to show the mechanism (e.g. what is the other receptor in SMCs; or how does NiV enter SMCs?).

Disucssion ll. 388 – 392: Authors state that a membrane-receptor such as ephrin-B2 traffics to the cell membrane after Fand G expression of NiV infection. Did they check for mRNA-levels of ephrin-B2 and/or ephrin B3 (see Figure 4)? Before and after F and G expression? Did they perform the FACS analysis with regards to surface expression of ephrin-B2 and ephrin-B3 after Fand G expression to look for differences? Since this further implies that NiV entry occurs independent from ephrins, did they try to inhibit or block endocytosis/micropinocytosis to see if SCMs are still infected? This all would help to unravel a claimed mechanism.

To assess productive SMCs infection, cell lysates for WB analysis are taken at the same point from cells infected with different MOI. This is not the correct wax to claim productive infection. Increase in NP detected could solely result from more input virus. For productive infection, one MOI and cell lysates collected over several time-points should be used.

With regards to the growth kinetics, please rephrase ll. 233-235. Titers after infection with a MOI of 5 peak up to 1x10^7 TCID50/ml, however, after 8 days of infection they are comparable to titers in the MOI 0.1 kinetic. Since medium is changed after day 6 on a regular basis, how do authors explain similar virus output between the two kinetics. Are the initially infected cells comparable and what about the infected cells at 21 DPI (SMCs)? Potentially, it would help to include a NiV-specific staining in Figure 2 A to show that SMCs are infected. Later, authors introduce HeLa cells that do not have ephrin-B2 or ephrin-B3. Are they infectable at a MOI 5 and result in NiV production?

Citation of Prescott et al., 2015 (25) for NiV F and G expression plasmids is odd since there is no description of expression plasmids in the cited study. Please correct and include a valid citation.

How often were experiments performed? N=x?

Presentation of data, especially the images in Figures 1, 3, 5 and 6, is poor. Since no magnifications for pictures are displayed, please include bars for size estimation. How should readers detect colocalization in Figure 1 of NiV NP and SM-actin in this magnification? Please use magnification insets. With regards to Figure 3, why is staining for NiV so weak at later timepoints for SMCs (compare 36 hr and 2.5 days to 16 and 18 hr). Also, cells need to be in contact with other cells to be able to fuse. SMCs look rather small and slim and the way sections are selected, one cannot see if they are indeed in contact with neighbouring cells. Maybe one could include pictures of light microscopy to see the cell layer and cell protrusions? Looking at Figure 5, please mark the syncytium at 24 hr since the picture is rather dark. With regards to the corresponding Figure legend, use the same words than in the Figure Mock – naïve and SMC – Mock. Figure 6 needs some thorough editing: In Figure 6 B the nuclei are difficult to detect – use an image with sharper contours. In Figures 6 C and D, the reader cannot detect the fused cells since the picture is rather blurry. If there are different cell layers, at least try to focus on the syncytium. Figure 6 E requires an inset to be able to see colocalization.

Overexpression of ephrin-B2 results in SMCs fusion (Figure 5). Is the same true for ephrin-B3 overexpression? After overexpression of ephrin-B2, can SMCs become as efficiently infected like ECs and infection kinetics become comparable to ECs? Also, Thiel et al., 2008 have shown that different amount of ephrin-B2 overexpressed have a negative effect on fusion and also that a high expression of ephrin-B2 down-regulates the NiV glycoproteins. Did authors test the impact of different amounts of ephrin-B2 overexpression? And how efficient is transfection of SMCs?

Please adjust the statement in l. 291: extensive fusion? I agree with fusion or advanced fusion, but extensive?

I have difficulties with Figure 6: First, as mentioned above, pictures are of poor quality and lack size bars. Second, in E authors claim that cells are mixed 1:1… I see a lot of GFP cells and fewer RFP cells. Third, how can authors be sure that after one SMCs fused with one ECs and thus passed F and G to ECs the following fusion events are not solely mediated by ECs and do not include SMCs (Figure 6D). Can co-staining with SMCs markers be performed? And lastly with regards to Figure 6 F, one syncytium is visible after 18 h and infection with MOI 5 and many non-fused ECs (green cells according to the labelling). In contrast in Figure 3, after 18 h and MOI 5 all ECs are fused and authors describe “complete destruction”. How is this disagreement in findings to exaplin?

Authors claim that NiV-induced cytotoxicity can occur in SMCs (l.292). I am sure the virus is not cytotoxic. The right term to use would be cytopathic effect/ cytopathogenicity.

Authors write several times using different wordings that “proteins were transfected” (e.g. ll. 305/306; 325/326). According to material and methods, expression plasmids were used for transfection. So, please correct the statements throughout the manuscript to be scientifically correct.

Figure Legend to Figure 6 (B): First of all, transfection of F alone will not lead to fusion of cells. G missing? And also, SMCs were not transformed but rather transduced with lentivirus.

  1. 390: “it is trafficked”… what do authors mean here? Receptor?

Discussion could be shorthened by leaving a repeat of results out.

Minor points:

It would be less confusing for the reader if authors would stick to either RFP or GFP labeling of ECs (Figure 6).

For easier reading of Figure 2 C, also include time-points in the upper graph.

Please be consistent in using or not using spaces between numbers and units (also in Figures). Please also decide if you want to abbreviate “hours” with h, hr or Hr.

Please be consistent with style of in vitro and in vivo (or in italics) throughout the manuscript.

Clearly, one heading for 3.6 would suffice?

  1. 322: better use “cell surface or cell membrane” instead of just membrane?

Unfortunately, there are many inconsistencies, grammatical errors and spelling mistakes throughout the manuscript indicating a certain carelessness; please correct:

  1. 23: …have not been reported?
  2. 26:… should it not read in this sentence: despite absence of cytopathic effects? Cytopathogenicity would be used when talking about the virus and not the cells?
  3. 27: contributors to what? Disease, disease progression, virus spread, pathogenicity,…
  4. 81: … and indicating… Sentence?

Ll. 127/128…, then complete…thereafter. Sentence?

Spelling of HeLa cells is “HeLa cells”. Please correct in Figure 4.

  1. 278: AF 647-conjugated
  2. 282: leads to NiV-induced fusion
  3. 293: possess
  4. 298: plasmids expressing
  5. 305: themselves….transfected…SMCs
  6. 308: Figure
  7. 313: derived
  8. 330: Brackets missing. … show
  9. 369: viral latency
  10. 387: expression: …only after transfection them…. F and G proteins of NiV…
  11. 394/395: Sentence? …at identifying…identified…?

Round 2

Reviewer 2 Report

DeByusscher et al., replied to the comments thoroughly and modified the manuscript accordingly. However, very few minor comments remain:

Figure 1: I still think that including insets (and not just increasing the whole figure) would help the reader to identify/see the relevant cells important for the authors conclusion.

l.79: detectable

l.273: undetectable

l. 290: conjugated

Figure 4, l. 286: I am sure this overlay just happened during insertion of the new figure

Author Response

We would again like to thank this reviewer for their time and effort. All changes were addressed with the exception of adding inserts to the figures. Inserts would only be enlarged areas of the same images that are presented. We now provide high resolution images that show the individual cell types.